# Q Fever in Unexplained Febrile Illness in Northern Algeria

H. Ghaoui [1,2,3,4,5,*], N. Achour [3,5,6], A. Saad-Djaballah [3,5,6], S. I. Belacel [3,5,6], I. Bitam [4,7,8] and P. E. Fournier [4]

1    IRD, MEPHI, Aix-Marseille Université, IHU Méditerranée Infection, 13005 Marseille, France
2    Ecole Nationale Supérieure Vétérinaire d'Alger, Rabie Bouchama, Alger 16059, Algeria
3    EHS des Maladies Infectieuses ELHADI FLICI, Alger 16022, Algeria; dr_barchiche@yahoo.fr (N.A.);
     asmaasaad@hotmail.fr (A.S.-D.); islembelacel30@gmail.com (S.I.B.)
4    IRD, APHM, VITROME, Aix-Marseille Université, 13005 Marseille, France; idirbitam@gmail.com (I.B.);
     pierre-edouard.fournier@univ-amu.fr (P.E.F.)
5    Association Scientifique Algerianne de Recherche en Infectiologie (ASARI), Alger 16047, Algeria
6    Faculté de Médecine, d'Alger Ziania-1-Université d'Alger 1 Benyoucef Benkhadda, Alger 16000, Algeria
7    Ecole Supérieure en Sciences de l'Aliment et des Industries Agroalimentaire d'Alger, Alger 16200, Algeria
8    Centre de Recherche en Agropastoralisme, Djelfa 17000, Algeria
*    Correspondence: ghaoui.hicham@hotmail.fr

**Abstract:** Our study aimed at assessing the role of *Coxiella burnetii* in nonspecific febrile illness at the National Center of Infectious Diseases in Algeria, EL-HADI FLICI Hospital. Seventy patients were included and compared to seventy controls without any ongoing infection. *Coxiella burnetii* infection was investigated using IFA serology and qPCR. Serology was positive in 3 of 70 patients (4.30%), including 1 in whom PCR was also positive (1.42%). All three patients reported frequent contact with ruminants. These results suggest that *C. burnetii* is an agent of nonspecific febrile illness in Algeria.

**Keywords:** *Coxiella burnetii*; Algeria; fever; IFA; qPCR

## 1. Introduction

*Coxiella burnetii* is the causative agent of Q fever, a worldwide disease that is mainly reported in Australia and various European countries, notably France [1]. Q fever is most often contracted after humans inhale infected dust particles or handle infected animal tissues, such as urine, feces or birth products [2]. Person-to-person transmission is exceptional [3].

Following inhalation, symptoms can develop after 10–90 days. This disease may be acute or chronic. Acute Q fever is asymptomatic in 60% of patients [4] but, when symptomatic, may present as an unexplained fever that makes diagnosis difficult for infectious disease specialists. When *C. burnetii* infection occurs, the onset is usually abrupt with high and prolonged fever, and may include myalgias, severe headache, cough and atypical pneumonia [4,5]. Though usually self-limited, acute Q fever may develop into a focalized (chronic) form [6]. Although the clinical spectrum of Q fever ranges from asymptomatic seroconversion to death (up to 1.5% of cases) [3], acute Q fever is most commonly mild. In chronic Q fever, cardiac valve (endocarditis) or vascular involvements make the disease potentially severe [3].

Q fever is often underdiagnosed and underreported because of its unspecific clinical signs, insufficient recognition by physicians and public health professionals, lack of awareness and limited diagnostic capabilities, especially in developing countries [7]. This disease has gained importance in the past two decades due the increasing number of reported cases and outbreaks [8,9]. Consequently, many organizations, such as the World Health Organization (WHO) and the United Nations (UN), defined it as an "Infection of Increasing Importance" (WHO 2004).

The immunofluorescence assay (IFA) is the reference method for the diagnosis of Q fever. It is based on the detection of antibodies against two antigenic phases of *C. burnetii* lipopolysaccharide: phase I and phase II antigens [10].

In Algeria, Q fever exists as an endemic disease that has been reported since 1948. However, since the outbreaks reported in Batna in 1957 and in Tlemcen in 1958 [11], there have been limited data on the prevalence of Q fever in humans in Algeria. In 1984, Dumas studied the seroprevalence of Q fever in Hoggar [12]. Then, Lachehab and Raoult [11] estimated the seroprevalence of Q fever in a population limited to eastern Algeria. In addition, in 2005, Benslimani et al. [13] studied infective endocarditis in Algeria and diagnosed two cases of Q fever. In addition, research efforts have been carried out in sheep, camels and ticks. The presence of *C. burnetii* was detected serologically in animal sera, and molecularly in ticks [14–18].

Given the nonspecific symptoms of Q fever and the lack of diagnostic tools in Algeria, infectious disease specialists in this country mostly rely on clinical features for the diagnosis of Q fever, which constitutes a diagnostic challenge. In consequence, there is no clear picture of the incidence of Q fever in Algeria. In light of these reasons, our study aimed at evaluating the role of *C. burnetii* in nonspecific febrile illness among patients hospitalized at the national center of infectious diseases: EL-HADI FLICI Hospital in Algiers.

## 2. Material and Methods

### 2.1. Study Design

In order to evaluate the role of Q fever as a cause of unexplained fever in Algeria, we performed our study at the National Reference Center for Infectious Diseases in Algeria in the ELHADI FLICI Hospital in Algiers, which admits patients from all Algerian departments. A case–control population-based study was conducted between July and October 2017. A total of 70 patients from different departments of northern Algeria were included in the study.

### 2.2. Inclusion Criteria and Cases Definitions

We only considered patients who were hospitalized for a nonspecific febrile illness associated or not with specific infectious syndromes such as hepatitis, meningitis, dermo-hypodermitis, sepsis, pneumonia and pyelonephritis. The presence of various clinical signs in favor of *C. burnetii* infection was investigated, including fever of unexplained origin with negative blood cultures, an acute respiratory syndrome, granulomatous hepatitis, influenza-like syndrome, chills, arthralgia, myalgia, purpuric or maculopapular skin rash and an undiagnosed infectious syndrome. These patients were included into a case group that was formed of 70 patients.

The control group included 70 patients who were admitted to the emergency ward of the hospital and were diagnosed with pathologies other than infectious diseases and had a normal blood cell count.

In parallel, for each patient (case group and control group), we completed a questionnaire including epidemiological data, housing area (rural or not), contact with animals, occupation and possibility of tick bite. Data were obtained through a standardized questionnaire to collect clinical information, contact with animals and health history. These data were analyzed retrospectively when the serological analysis or molecular test was positive.

### 2.3. Ethics Statement

The study design was validated by the ethics committee of the EL-HADI FLICI Hospital prior to starting the study. All patients gave their informed consent to be included in the study and have their interview information published, and they agreed to have a blood sample taken.

### 2.4. Sample Collection

A total of 140 samples, comprising whole blood/sera from each patient (control and case groups), were collected. Samples were collected aseptically, the sera in dry tubes and the whole blood in EDTA tubes (EDTA:Ethylenediaminetetraacetic acid). The samples were

conserved at −20 °C and transferred to the Mediterranee Infection Institute, Marseille, for IFA serology and *C. burnetii* qPCR.

### 2.5. Serological Assays

Serologic tests were performed using an indirect immunofluorescence assay (IFA), which is the reference method for the serodiagnosis of Q fever. We used the reference strain *C. burnetii* Nine Mile as an antigen. Antigen preparation and purification were performed as previously described [19].

### 2.6. DNA Extraction and Real-Time PCR

A total of 200 µL of DNA was extracted from each EDTA sample using the QIAamp Tissue Kit and the QIAGEN-BioRobot EZ1, according to the manufacturer's instructions (Qiagen, Hilden, Germany). Extracted DNA was stored at −20 °C in sterile conditions until it was used for PCR.q PCR was used for the detection of *C. burnetii* by employing specific primers and a probe designed to amplify the IS1111 gene and confirmed with the second gene, IS30a, which remains highly *C. burnetii*-species specific. We added 5 µL of DNA and 15 µL of mix from the Roche PCR Kit (Roche Diagnostics, Meylan, France). The qPCR cycling parameters were 5 min at 95 °C followed by 39 cycles, each consisting of 5 s of denaturation at 95 °C and 30 s of annealing at 60 °C.

### 2.7. Statistical Analyses

In order to calculate the significance level (*p*-values) of the various results obtained, we applied Yates correction for the chi-square test. *p*-values lower than 0.05 were considered statistically significant.

## 3. Results
### 3.1. Patient Characteristics

The mean age of the study population was $36 \pm 18$ years old, with patients ranging from 5 to 72 years. Sixty-three were men and seventy-seven were women.

### 3.2. IFA Serology

Using IFA, all sera belonging to control group patients were found to be negative for antibodies to *C. burnetii*, whereas 3/70 case group members (4.28%) were positive (Table 1), with 2 exhibiting positive IgG titers to phase II and one showing both IgG and IgM titters. These results suggest that one patient had an active acute Q fever and two had evidence of past Q fever.

**Table 1.** IS30a q PCR results for the control and case groups.

| Groups | Positive Sera | *Coxiella burnetii* | | | IS1111/ IS30a q PCR |
|---|---|---|---|---|---|
| | | IgG | IgM | IgA | |
| Control group | 0/70 | - | - | - | 0/70 |
| Case group | 3/70 * NS *p* = 0.258 (4.30%) | | | | 1/70 (1.43%) |
| | N°1 | 100 | 0 | 0 | n |
| | N°2 | 100 | 0 | 0 | n |
| | N°3 | 200 | 100 | 200 | Ct 33.21/33.74 |

* n: negative. * All sera were first screened with total immunoglobulin. If the serum was positive at 1/100 dilution, then the antibodies present in this sample were considered differentially quantified (IgG, IgM, IgA). * significance (S) was determined when the *p*-value was ≤0.05. NS: no significant.

### 3.3. Detection of Coxiella burnetii via qPCR

Only one whole blood sample (1.4%) was positive for both IS1111 and IS30a targets in the case group. In contrast, all control group samples were negative (Table 1). The Ct values of the positive sample were 33.21 and 33.74 for IS111 and IS30a, respectively (corresponding to 4.8 and 4.7 log10 DNA copies/mL), and the Cts of the positive control were 26.58 and 27.02 for IS1111 and IS30a, respectively (corresponding to 6.8 and 6.7 log10 DNA copies/mL). The PCR-positive patient was also the one exhibiting an acute Q fever serology profile. However, the other two IFA-positive patients were PCR-negative. Overall, 3/70 patients in the case group but none in the control group exhibited evidence of *C. burnetii* infection. The difference between the two groups was not statistically significant ($p = 0.258$).

### 3.4. Positive Case Description

The three positive patients presented with different clinical signs that are summarized in Table 2. In addition to unexplained fever, two patients were diagnosed with meningeal syndrome. The third patient presented arthro-myalgia pains, maculopapular rash without pruritus and hepatic hilar adenopathies detected via a thoracoabdominal CT scan. This patient also exhibited hyperleukocytosis, a thrombocytopenia value of 89,000/mm$^3$, a CRP level of 42 mg/L and hepatic cytolysis with transaminase levels 5 times higher than normal. This disturbed biological statute may explain the underlying infection with *C. burnetii*, in addition to the visceral leishmaniosis diagnosed previously when the patient was admitted in Algeria.

**Table 2.** Epidemiologic data, clinical manifestations and paraclinical findings of the positive *Coxiella burnetii* cases.

|  | First Case<br>CB IFA+/qPCR- | Second Case<br>CB IFA+/qPCR- | Third Case<br>CB IFA+/qPCR+ |
|---|---|---|---|
| Age/Sex | 8-year-old female | 18-year-old female | 19-year-old female |
| Fever | Prolonged unexplained fever | Prolonged unexplained fever | Prolonged unexplained fever |
| Diagnosis | Meningeal syndrome with brutal installation for 2 days | Meningeal syndrome | Visceral leishmaniosis |
| Clinical signs | Arthralgia<br>Large joints<br>Purpuric rash<br>Asthenia | Nothing special | Significant myalgia syndrome<br>Maculopapular rash<br>without pruritus |
| Paraclinical findings | - Cerebrospinal fluid clear, 1800/mm$^3$<br>- Cerebrospinal fluid Albumin = 0.52 g/L<br>- Cerebrospinal fluid Normal glucose level: 0.48 g/L<br>CRP < 6 | - Cerebrospinal fluid clear, 700/mm$^3$<br>- Cerebrospinal fluid Albumin = 0.94/L<br>- Cerebrospinal fluid Normal glucose level: 0.43 g/L.<br>CRP < 6<br>Normal cerebral CT-scan | Complete blood count with hyperleucytosis (29,000/µL), CRP 42 mg/L<br>Thrombocytopenia at 89,000/mm$^3$<br>Hepatic cytolysis:<br>AST = 130 UI/L;<br>ALT = 125 UI/L |
| Contact with animals | Cats, sheep, dog | Sheep, cattle | Goat, sheep, camels, hare |
| Evolution | Full recovery | Moderate recovery | Unknow evolution (exited hospital against medical advice) |

## 4. Discussion

Northern Algeria is a hot and humid region where vector-borne infectious diseases occur mostly during the summer season. Q fever is an anthropozoonosis that has lacked attention, in terms of studies and epidemiological surveillance, in Algeria. Apart from the reports of outbreaks in 1948, 1957 and 1958 [11], scarce data are available on the disease [11,12].

We aimed at evaluating the burden of Q fever among patients hospitalized for unexplained fever at the National Center of Infectious Diseases in Algiers, ELHADI FLICI Hospital.

Among the 140 patients investigated between July and October 2017, the seroprevalence of Q fever among the case group was 4.3%, including one patient who exhibited an antibody profile of acute infection and was also PCR-positive. These results support the diagnosis of active acute Q fever in this patient. The other two seropositive patients had serology profiles consistent with past infection.

Despite being distributed worldwide, the seroprevalence of *C. burnetii* may differ according to regions and occupations [20,21]. In Algeria, a study performed in 1978 in Hoggar, in the southern part of the country, determined that the overall seroprevalence in humans was 5.4%, with a majority of seropositive patients being males younger than 16 years old [12]. This study also described seropositive sheep, goats and camels in Hoggar. In 1996, a study performed in the Setif region (north-eastern Algeria) disclosed a seroprevalence of 15.5%, confirming the endemic presence of Q fever in this area [11]. In the same study, a similar seroprevalence of 14.2% was reported in the Aures region (also north-eastern Algeria) [11]. This study was conducted in slaughterhouse workers from Algiers, and found a 15% seroprevalence in humans. There are no data on Q-fever seroprevalence humans in other Algerian regions. However, in Tunisia, the eastern neighboring country, the disease is described as endemic [22], with seroprevalences of 26% in blood donors [23] and 5.88% in febrile patients [24]. In Morocco, Algeria's western neighbor, seroprevalences of 1% and 18.3% were reported in Casablanca and Fez, respectively [25]. These data suggest that the disease is widely present throughout Maghreb.

In our study, the seroprevalence in patients with unexplained fever was 4.3%. All three seropositive patients had in common frequent contact with ruminants. One patient, from southern Algeria, lives a nomadic lifestyle; the other two come from the suburbs of Algiers (being companions) and are owners of sheep farms. Algeria hosts many cattle, goats and sheep farms, which remain major sources of *Coxiella burnetii* infection. Many studies have been conducted on *C. burnetii* in animals in Algeria. In 2016, a seroprevalence of 14.1% was observed in small ruminant flocks, with a vaginal carriage positivity of 21.3% [15]. Other studies reported the presence of *C. burnetii* in 0.8% of dog spleens [14], in 5.8 and 1.7% of sheep and goat blood specimens, respectively, and in 5.5% of ticks collected from ruminants [16]. Although ticks play a minor role in the transmission of Q fever to humans, their role in its transmission among animals should not be underestimated. In 2022, in the east province of Algeria, a serological and molecular study on its prevalence in bulk tank milk (BTM) was conducted; the results highlighted 37% *Coxiella burnetii* antibody (using indirect enzyme–linked immunosorbent assay) and 9% *Coxiella burnetii* DNA in the BTM samples [18]. In addition, camels have been demonstrated to be potential sources of *C. burnetii* [17]. In this respect, 75.5% seroprevalence of antibodies against *C. burnetii* was detected in camels in Algeria in 2020 using the ELISA kit, while detection of *C. burnetii* DNA in ticks from camels showed a prevalence of 11.66% [18]. The patient with acute Q fever in our study also reported contact with camels in addition to goats and sheep. The period of our study also coincided with the Aid EL-AD'HA, a Muslim religious feast during which Algerians may be in close contact with sheep. These results highlight the various sources of *C. burnetii* contamination in Algeria.

In light of the results obtained in this study and the literature, we emphasize that human Q fever is endemic in Algeria, where it remains a public health problem and should be considered among the differential diagnoses of nonspecific febrile illnesses. It is also necessary to raise the awareness of the disease in ruminant farmers and mobilize public health actors to better identify the sources of contamination and excretion routes of *C. burnetii*. Further studies focused on genotyping animal and human strains are also suggested to better understand the epidemiological role of farming animals in Algeria.

**Author Contributions:** H.G. and N.A. conceptually designed the strategy for this study and drafted the manuscript; N.A., H.G., A.S.-D. and S.I.B. ensured the diagnosis at hospital, sampling and follow-up of patients; H.G. analyzed the samples in the laboratory; I.B. analyzed the data and drafted the manuscript; P.E.F. analyzed the data and corrected the manuscript and participated in discussion. All authors have read and agreed to the published version of the manuscript.

**Funding:** The Agence Nationale pour la Recherche (Programme d'investissement d'Avenir, Reference Méditerranée Infection 10-IAHU-03).

**Institutional Review Board Statement:** Not applicable.

**Informed Consent Statement:** Not applicable.

**Data Availability Statement:** Specific data can be made available upon request.

**Acknowledgments:** The authors thank the medical and the technical staff of the French Reference Center for Q fever: IHU, Marseille, France.

**Conflicts of Interest:** The authors declare no competing interests in relation to this research.

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
