# Peer review of "Q Fever in Unexplained Febrile Illness in Northern Algeria"

_2036-7481, doi:10.3390/microbiolres14040109_

Round 1

Reviewer 1 Report

Dear Editor,

In this article, the authors deal with assessing the role of Coxiella burnetii in nonspecific febrile illness at the National Center of Infectious Diseases in Algeria, respectively EL-HADI FLICI Hospital. The idea of the manuscript is very good, but there are still some points to be addressed by the authors in order to improve their manuscript.

Introduction section

 Line 33 - I recommend to check spelling

Material and Methods

The number of patients included in the case group is relatively small, especially when taking into account the low prevalence of the previously detected infection.

I recommend adding more information about the primers used for the detection of Coxiella spp.

Discussion section

Add a few references related to Q fever in Algeria:

DOI: 10.1016/j.actatropica.2019.02.032

DOI: 10.3390/vetsci9020040.

DOI: 10.1016/j.actatropica.2020.105443

Line 149 - I recommend to check spelling

Thank you for considering me as a reviewer, and I am open to dialogue to support the development and publication of a high-quality article.

Best regards,

Reviewer 2 Report

Dear authors,

the topic is becoming more and more interesting for clinicians. The paper is well written, results and conclusions are clear.  Below are my specific comments:

Use Coxiella burnetii ecxtended only the first time, or at the beginning of paragraph, then abbreviate C. burnetii along the whole manuscript.

L55 Please cite new recent studies  in Algeria e.g. Menadi et al. 2020 and 2022 on serological and molecular investigation in cattle (blood and milk)

L104 200 μl avoid bold

L108 is1111 in capital letter IS1111

L108 Please specify which kit for C. burnetii has been used or which primers and probe you used for IS1111 amplification

L112 please avoid the space and modify to IS30a. Moreover, as for IS1111 detail all the reagents used for this specific qPCR in order to share reproducible protocols

L146 check for double space

L172-174 Discuss here data from cattle seroprevalence reported by Menadi et al. 2020 and blood/milk PCR positive samples (Menadi 2022). In the last paper it has been also shown a small picture of the circulating genotypes in cows, thus it will be very interesting to genotype human strains in order to understand the cow's role in this area for human Q Fever

L176 Use capital letter

L177 Add "in" seroprevalens in humans

L199-204 These results are interesting, but Q fever is still not considered a 'major' public problem. I suggest removing 'major' and rephrase as follows:  Q fever is endemic in Algeria where it remains a public health problem and should be considered...
Moreover, the adjective major is in contrast to the non-statistically significant difference between the control and test groups

L201 Illness in lowercase

L205 Further studies focused on genotyping of animal and human strains is also suggested to better understand the epidemiogical role of farming animals in Algeria
